# CLASSIFICATION WITH CONCEPTUAL SAFEGUARDS

**Hailey Joren**
UC San Diego
hjoren@ucsd.edu

**Charles Marx**
Stanford University
ctmarx@stanford.edu

**Berk Ustun**
UC San Diego
berk@ucsd.edu

## ABSTRACT

We propose a new approach to promote safety in classification tasks with established concepts. Our approach – called a *conceptual safeguard* – acts as a verification layer for models that predict a target outcome by first predicting the presence of intermediate concepts. Given this architecture, a safeguard ensures that a model meets a minimal level of accuracy by abstaining from uncertain predictions. In contrast to a standard selective classifier, a safeguard provides an avenue to improve coverage by allowing a human to confirm the presence of uncertain concepts on instances on which it abstains. We develop methods to build safeguards that maximize coverage without compromising safety, namely techniques to propagate the uncertainty in concept predictions and to flag salient concepts for human review. We benchmark our approach on a collection of real-world and synthetic datasets, showing that it can improve performance and coverage in deep learning tasks.

## 1 INTRODUCTION

One of the most promising applications of machine learning is to automate routine tasks that a human can perform. We can now train deep learning models to perform such tasks across applications – be it to identify a bird in an image [1], detect toxicity in text [2], or pneumonia in a chest x-ray [3]. Even as these models may outperform human experts [4, 5, 6], their performance falls short of the levels we need to reap the benefits of full automation. A bird identification model that is 80% accurate may not work reliably enough to be used in the field by conservationists. A pneumonia detection model with 95% accuracy is still not sufficient to eliminate the need for human oversight in a hospital setting.

One strategy to reap some of the benefits of automation in such tasks is through *abstention*. Given any model that is insufficiently accurate, we can measure the uncertainty in its predictions and improve its accuracy by abstaining from predictions that are too uncertain. In this way, we can improve accuracy by sacrificing *coverage* – i.e., the proportion of instances where a model assigns a prediction. One of the common barriers to abstention in general applications is how to handle instances where a model abstains. In applications where we wish to automate a routine task, we would only pass these to the human expert who would have made the decision in the first place. Thus, abstention represents a way to reap benefits from *partial automation*.

In this paper, we present a modeling paradigm to ensure safety in such applications that we call a *conceptual safeguard* (see Fig. 1). A conceptual safeguard is an abstention mechanism for models that predict an outcome by first predicting a set of concepts. Given such a model, a conceptual safeguard operates as a verification layer – estimating the uncertainty in each prediction and abstaining when it exceeds a threshold needed to ensure a minimal level of accuracy. Unlike a traditional selective classifier, a conceptual safeguard provides a way to improve coverage – by allowing experts to confirm the presence of certain concepts on instances on which we abstain.

Although conceptual safeguards are designed to be a simple component that can be implemented off the shelf, designing methods to learn them is challenging. On the one hand, concepts introduce a degree of uncertainty that we must account for at test time. In the best case, we may have to abstain too much to achieve a desired level of accuracy. In the worst case, we may fail to hit the mark. On the other hand, we must build systems that are *designed for confirmation* – i.e., where we can reasonably expect that confirming concepts will improve coverage and where we can rank the concepts in a way that improves coverage. Our work seeks to address these challenges so that we can reap the benefits of these models.

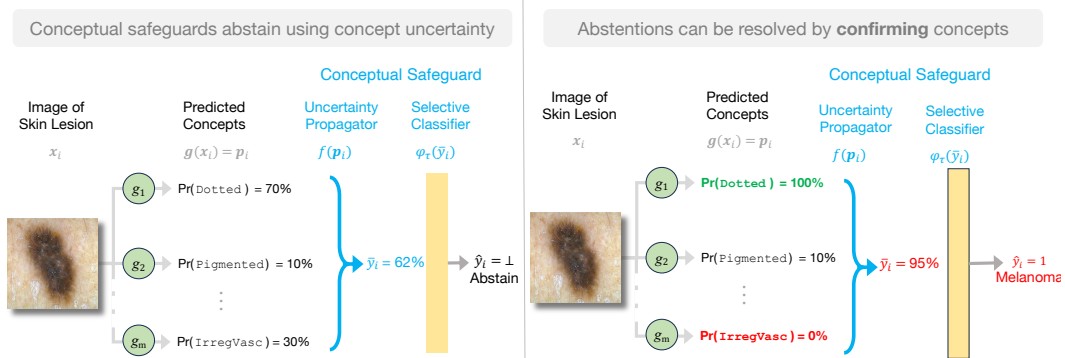

**Figure 1:** Conceptual safeguard to detect melanoma from an image of a skin lesion. We consider a model that estimates the probabilities of $m$ concepts: `Dotted`, `Pigmented`...`IrregularVasc`. Given these probabilities, a conceptual safeguard will decide whether to output a prediction $\hat{y} \in \{$`Melanoma`, `NoMelanoma`$\}$ or to abstain $\hat{y} = \perp$. The safeguard improves accuracy by abstaining on images that would receive a low confidence prediction, and measures confidence in a way that accounts for the uncertainty in concept predictions through uncertainty propagation. On the left, we show an image where a safeguard abstains because its confidence fails to meet the threshold to ensure high accuracy $\Pr(\texttt{Melanoma}) = 62\% \leq 90\%$. On the right, we show a human expert can resolve the abstention by confirming the presence of concepts `Dotted` and `IrregVasc` in the image.

**Contributions**     Our main contributions include:

1. We introduce a general-purpose approach for safe automation in classification tasks with concept annotation. Our approach can be applied using off-the-shelf supervised learning techniques.

2. We develop a technique to account for concept uncertainty in concept bottleneck models. Our technique can use native estimates from the components of a concept bottleneck to return reliable estimates of label uncertainty, improving the accuracy-coverage trade-off in selective classification.

3. We propose a confirmation policy that can improve coverage. Our policy flags concepts in instances on which a model abstains, prioritizing high-value concepts that could resolve abstention. This strategy can be readily customized to conform to a budget and applied offline.

4. We benchmark conceptual safeguards on classification datasets with concept labels. Our results show that safeguards can improve accuracy and coverage through uncertainty propagation and concept confirmation.

RELATED WORK

**Selective Classification**     Our work is related to a large body of work on machine learning with a reject option [see e.g., 7, for a recent survey], and more specifically methods for selective classification [8, 9, 10, 11]. Our goal is to learn a selective classifier that assigns as many predictions as possible while adhering to a minimal level of accuracy [i.e., the bounded abstention model of 11]. We build conceptual safeguards for this task through a *post-hoc* approach [see e.g., 12] – in which we are given a model, and build a verification layer that estimates the confidence of each prediction and abstains on predictions where a model is insufficiently confident. The key challenge in our approach is that we require reliable estimates of uncertainty to abstain effectively [13, 14], which we address by propagating the uncertainty in concept predictions to the uncertainty in the final outcome.

**Deep Learning with Concepts**     Our work is related to a stream of work on deep learning with concept annotations. A large body of work uses these annotations to promote interpretability – either by explaining the predictions of DNN in terms of concepts that a human can understand [15, 16], or by building *concept bottleneck models* – i.e., a model that predicts an outcome of interest by predicting concepts – i.e., [17]. One of the key motivations for concept bottleneck models is the potential for humans to intervene at test time [17] – e.g., to improve performance by correcting a concept prediction that was incorrectly detected. Recent work shows that many architectures that

perform well cannot readily support interventions [see e.g., 18, 19].[1] Likewise, interventions may not be practical in applications where we wish to automate a routine task. In such cases, we need a mechanism to flag predictions for human review to prevent human experts from having to check each prediction. Our work highlights several avenues to avoid these limitations. In particular, we work with an independent architecture that is amenable to interventions, consider a restricted class of interventions, and present an approach that does not require constant human supervision.

One overarching challenge in building concept bottleneck models is the need for concept annotations. In effect, very few datasets include concept annotations and those that do are often incomplete (e.g., an example may be missing some or all concept labels). In practice, the lack of concept labels can limit the applicability and the performance of concept bottleneck models – and has motivated work a recent stream of work on machine-driven concept annotation [see e.g., 20] and drawing on alternate sources of information [21, 22]. Our work outlines an alternative approach to overcome this barrier to adoption: rather than building a new model that is sufficiently accurate, use it as much as possible by allowing it to abstain from prediction.

## 2 FRAMEWORK

We consider a classification task to predict a label from a complex feature space. We start with a dataset of $n$ i.i.d. training examples $\{(\mathbf{x}_i, \mathbf{c}_i, y_i)\}_{i=1}^n$, where example $i$ consists of:

- a vector of *features* $\mathbf{x}_i \in \mathcal{X} \subseteq \mathbb{R}^d$ – e.g., $\mathbf{x}_i$ pixels in image $i$;
- a vector of $k$ *concepts* $\mathbf{c}_i \in \mathcal{C} = \{0,1\}^m$ – e.g., $c_{i,k} = 1$ if x-ray $i$ contains a bone spur;
- a *label* $y_i \in \mathcal{Y} = \{0,1\}$ – e.g., $y_i = 1$ if patient $i$ has arthritis.

**Objective** We use the dataset to build a selective classification model $h : \mathcal{X} \to \mathcal{Y} \cup \{\perp\}$. Given a feature vector $\mathbf{x}_i$, we denote the output of the model as $\hat{y}_i := h(\mathbf{x}_i)$ where $\hat{y}_i = \perp$ denotes that the model *abstains from prediction* for $\boldsymbol{x}_i$.

In the context of an automation task, we would like our model to assign as many predictions as possible while adhering to a target accuracy to ensure safety at test time. In this setup, abstention reflects a viable path to meet this constraint – by allowing the model to abstain on instances where it would otherwise assign an incorrect prediction. Given an *target accuracy* $\alpha \in (0,1)$, we express these requirements as an optimization problem:

$$\max_h \quad \text{Coverage}(h)$$
$$\text{s.t.} \quad \text{Accuracy}(h) \geq \alpha,$$

where:

- $\text{Coverage}(h) := \Pr(\hat{y} \neq \perp)$ is the *coverage* of the model $h$ – i.e., proportion of instances where $h$ outputs a prediction.
- $\text{Accuracy}(h) := \Pr(y = \hat{y} \mid \hat{y} \neq \perp)$ is the *selective accuracy* of the model $h$ – i.e., the accuracy of $h$ over instances on which it outputs a prediction

**System Components** We consider a selective classifier with the components shown in Fig. 1. Given the dataset, we will first train the basic components of an independent concept bottleneck model:

- A *concept detector* $g : \mathcal{X} \to [0,1]^m$. Given features $\boldsymbol{x}_i$, the concept detector returns as output a vector of $m$ probabilities $\mathbf{q}_i = [q_{i,1}, \ldots, q_{i,m}] \in [0,1]^m$ where each $q_{i,k} := \Pr(c_{i,k} = 1 \mid \boldsymbol{x}_i)$ captures the probability that concept $k$ is present in instance $i$.

- A *front-end model* $f : \mathcal{C} \to \mathcal{Y}$, which takes as input a vector of *hard* concepts $\mathbf{c}_i$ and returns as output the outcome probability $\overline{y}_i := f(\mathbf{c}_i)$.

---

[1]For example, if we build a sequential architecture – wherein we train a front-end model to predict the label from *predicted concepts* – then interventions may reduce accuracy as the front-end may rely on an incorrect concept prediction to output accurate label predictions.

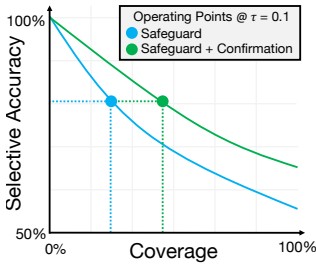

**Figure 2:** We evaluate the performance of classification models that can abstain using an accuracy-coverage curve [11]. Given a model that outputs a probability prediction for each point, a conceptual safeguard flags points on which a model abstains based on a confidence threshold $\tau \in [0, 0.5]$ – where setting $\tau = 0$ leads to 100% coverage and setting $\tau = 0.5$ leads to 0% coverage.

In what follows, we treat these models as fixed components that are given and assume that they satisfy two requirements: (i) *independent training*, meaning that we train the concept detector and the front-end model using the datasets $\{(\mathbf{x}_i, \mathbf{c}_i)\}_{i=1}^n$ and $\{(\mathbf{c}_i, y_i)\}_{i=1}^n$, respectively; (ii) *calibration*, i.e., that the concept detector and front-end model and return calibrated probability estimates of their respective outcomes. As we will discuss shortly, independent training will be essential for confirmation while calibration will be essential for abstention.

In practice, these are mild requirements that we can satisfy using off-the-shelf methods for supervised learning. Given a dataset, for example, we can fit the concept detectors using a standard deep learning algorithm, and fit the front-end model using logistic regression. In settings where the dataset is missing concept labels for certain examples, we can train a separate detector for each concept to maximize the amount of data for each concept detector. In settings where we have an embedding model [21] or an existing end-to-end deep learning model, we can dramatically reduce the computation by training the concept detectors via fine-tuning. In all cases, we can calibrate each model by applying a post-hoc calibration technique, e.g., Platt scaling [23].

**Conceptual Safeguards** Conceptual safeguards operate as a confidence-based selective classifier – abstaining on points where they cannot assign sufficiently confident prediction. We control this behavior through an internal component called the *selection gate* $\varphi_\tau : [0, 1] \to \mathcal{Y} \cup \{\perp\}$, which is parameterized with a *confidence threshold* $\tau \in [0, 1]$. Given a threshold $\tau$, the gate takes as input a soft label $\overline{y}_i \in [0, 1]$ and returns:

$$\hat{y}_i = \varphi_\tau(\overline{y}_i) = \begin{cases} 0 & \text{if} \quad \overline{y}_i \in [0, \tau) \\ \perp & \text{if} \quad \overline{y}_i \in [\tau, 1 - \tau] \\ 1 & \text{if} \quad \overline{y}_i \in (1 - \tau, 1] \end{cases}$$

As shown in Fig. 2, we can set $\tau$ to choose an operating point for the model on the accuracy-coverage curve – e.g., to meet a desired target accuracy rate at by sacrificing coverage. In settings where our front-end model returns calibrated probability predictions, we can use them to set the threshold $\tau$:

**Definition 1.** A probabilistic prediction $\overline{y} \in [0, 1]$ for a binary label $y \in \{0, 1\}$ is *calibrated* if $\Pr(y = 1 \mid \overline{y} = t) = t$ for all values $t \in [0, 1]$.

**Proposition 1.** *Suppose that $\overline{y}$ is a calibrated probability prediction for $y$. Then any selective classifier $\varphi_\tau(\overline{y})$ that abstains when $\overline{y}$ has confidence below $1 - \tau$ achieves accuracy at least $1 - \tau$.*

Proposition 1 is a standard result ensuring high accuracy for selective classifiers that output calibrated probabilistic prediction (see Appendix A for a proof). In cases where the front-end model is not calibrated, the result will hold only given the degree of calibration. Thus, the reliability of this approach hinges on the reliability of our confidence estimates. An alternative approach for such cases is to treat the output of the front-end model $f$ as a *confidence score* and tune $\tau$ using a calibration dataset [see, e.g., the SGR algorithm of 12].

**Confirmation** We let users *confirm* the presence of concepts on instances where a model abstains. In a pneumonia detection task, for example, we can ask a radiologist to confirm the presence of

concept $k$ in x-ray $i$. We refer to this procedure as confirmation rather than intervention to distinguish it from other ways where a human expert would alter the output from a concept detector.[2]

We consider tasks where each concept is *human-verifiable* – i.e., that it can be detected correctly by the human experts that we would query to confirm [c.f., concepts where a human may be uncertain as in 24]. In such tasks, confirming a concept will replace its probability $q_{i,k}$ to its ground-truth value $c_{i,k} \in \{0, 1\}$. We cast confirmation as a *policy* function $\psi_S : [0, 1]^m \to [0, 1]^m$ where $S \subseteq [m]$ denotes a subset of concepts to confirm. The function takes as input a vector of raw concept predictions and returns a vector of partially confirmed concept predictions: $\boldsymbol{p}_i = [p_{i,1}, \ldots, p_{i,m}] \in [0, 1]^m$ where:

$$p_{i,k} := \begin{cases} c_{i,k} & \text{if } k \in S \\ q_{i,k} & \text{if } k \notin S \end{cases}$$

## 3 METHODOLOGY

In this section, we describe how to build the internal components of a conceptual safeguard. We first discuss how to output reliable predicted probabilities given uncertain inputs at test time. We then introduce a technique to flag promising examples that can be confirmed to improve coverage.

### 3.1 UNCERTAINTY PROPAGATION

It is important that the concept detectors are probabilistic, so that we can prioritize confirming concepts that have high uncertainty. However, this creates an issue where the concept detectors output probabilities, but the front-end model requires hard concepts as inputs.

Here, we describe a simple strategy wherein we use *uncertainty propagation* to allow the front-end model to accept probabilities rather than hard concepts as input. We make two assumptions about the underlying data distribution to motivate our approach.

**Assumption 2.** *The label $y$ and features $\boldsymbol{x}$ are conditionally independent given the concepts $\boldsymbol{c}$.*

**Assumption 3.** *The concepts $\{c_1, \ldots, c_m\}$ are conditionally independent given the features $\boldsymbol{x}$.*

We can use these assumptions to write the conditional label distribution $p(y \mid \boldsymbol{x})$ in terms of quantities that we can readily obtain from a concept detector and front-end model:

$$p(y \mid \boldsymbol{x}) = \sum_{\boldsymbol{c} \in \{0,1\}^m} p(y \mid \boldsymbol{c}, \boldsymbol{x}) \, p(\boldsymbol{c} \mid \mathbf{x}) \tag{1}$$

$$= \sum_{\boldsymbol{c} \in \{0,1\}^m} p(y \mid \boldsymbol{c}) \, p(\boldsymbol{c} \mid \mathbf{x}) \tag{Assumption 2}$$

$$= \sum_{\boldsymbol{c} \in \{0,1\}^m} \underbrace{p(y \mid \boldsymbol{c})}_{\text{output from } f(\mathbf{c})} \prod_{k \in [m]} \underbrace{p(c_k \mid \boldsymbol{x})}_{\text{output from } g_k(\mathbf{x})} \tag{Assumption 3}$$

We use this decomposition to propagate uncertainty from the inputs of the front-end model to its output. Specifically, we compute the expected prediction of the front-end model on all possible realizations of hard concepts and weigh each realization in terms of the probabilities from concept detectors. In practice, we replace each quantity in Eq. (1) with an estimate computed by the concept detectors and front-end model. Given predicted probabilities from concept detectors $\boldsymbol{p}_i = (p_{i,1}, \ldots, p_{i,k})$, we compute the estimate of $p(y \mid \boldsymbol{x})$ as:

$$f(\boldsymbol{p}_i) := \sum_{\boldsymbol{c} \in \{0,1\}^m} f(\boldsymbol{c}) \prod_{k \in [m]} p_{i,k}^{c_k} (1 - p_{i,k})^{1 - c_k} \tag{2}$$

Here, we abuse notation slightly and use $f(\boldsymbol{p}_i)$ to denote the front-end model applied to uncertain concepts, and $f(\boldsymbol{c})$ to denote the front-end model applied to hard concepts. This requires $2^m$ calls to the front-end model, which is negligible in practice as most front-end models are trained with a limited number of concepts. In settings where $m$ is large, or computation is prohibitive, we can use a sample of concept vectors to construct a Monte Carlo estimate.

---

[2]For example, intervention may refer to "correction" in which a human replaces a hard concept prediction with its correct value.

---

**Algorithm 1** Greedy Concept Selection

---

**Input:** $\{i \in [n] \mid \varphi_\tau(f(\mathbf{q}_i)) = \bot\}$ instances on which a model abstained
**Input:** $\gamma_1, \ldots, \gamma_m > 0$, cost to confirm each concept
**Input:** $B > 0$, confirmation budget
1:  $S_1, \ldots, S_n \leftarrow \{\}$                                          *concepts to confirm for each instance*
2:  **repeat**
      $i^*, k^* \leftarrow \arg\max_{i,k} \ \mathrm{Gain}(\mathbf{q}_i, k)$ s.t. $k \notin S_i$               *select best remaining concept*
3:      $S_{i^*} \leftarrow S_{i^*} \cup \{k^*\}$
4:      $B \leftarrow B - \gamma_{k^*}$
5:  **until** $B < 0$ or $S_i = [m]$ for all $i \in [n]$
**Output:** $S_1, \ldots, S_n$, concepts to confirm for each abstained instance

---

## 3.2 CONFIRMATION

Selective classification guarantees higher accuracy at the cost of potentially lower coverage. In what follows, we describe a strategy that can mitigate the loss in confirmation by human confirmation – i.e., manually spotting concepts among instances on which we abstain. In principle, confirmation will always lead to an improvement in coverage. In practice, human confirmation is labor intensive – and may require expertise – so we want to develop techniques that can account by flagging promising examples that are responsive to confirmation costs.

In Algorithm 1, we present a routine to flag concepts for a human expert to review among instances on which a model abstains. The routine iterates over a batch of $n$ points on which the model has abstained and identifies salient concepts that can be confirmed by a human expert to avoid coverage.

Algorithm 1 computes the gain associated with confirming each concept on each instance and then returns the concepts with the maximum gain while adhering to a user-specified *confirmation budget* $B > 0$. We associate the cost of confirming each concept $k$ with a cost $\gamma_k > 0$, which can be set to control the time or expertise that is associated with each confirmation. The routine selects concepts for review based on the expectation of the gain in certainty. We measure the gain in certainty in terms of the variance of the prediction after confirmation:

$$\begin{aligned}
\mathrm{Gain}(\mathbf{q}, k) &= \mathrm{Var}_{p_k \sim \mathrm{Bern}(q_k)} \left[ f(\psi_{\{k\}}(\mathbf{q})) \right] \\
&= q_k(1 - q_k) \left( f(\mathbf{q}[q_k \leftarrow 1]) - f(\mathbf{q}[q_k \leftarrow 0]) \right)^2
\end{aligned} \tag{3}$$

The gain measure in (3) captures the sensitivity of predictions from the front-end model by confirming concept $k$. In particular, we seek to identify concepts that – if confirmed – would induce a large change in the output of the front-end and thus resolve abstentions. Given that we do not know the underlying value of concept $k$ prior to confirmation, we treat $q_k$ as a random variable that will be set to $\mathbf{q}[q_k \leftarrow 1]$ with probability $q_k$ and $\mathbf{q}[q_k \leftarrow 0]$ with probability $1 - q_k$. Here, $\mathbf{q}[q_k \leftarrow 1]$ refer to the vector $\mathbf{q}$, except with $q_k$ replaced with 1.

## 4 EXPERIMENTS

We present experiments where we benchmark conceptual safeguards on a collection of real-world classification datasets. Our goal is to evaluate their accuracy and coverage trade-offs, and to study the effect of uncertainty propagation and confirmation through ablation studies. We include details on our setup and results in Appendix B, and provide code to reproduce our results on GitHub.

## 4.1 SETUP

We consider six classification datasets with concept annotations:

- The `melanoma` and `skincancer` datasets are image classification tasks to diagnose melanoma and skin cancer derived from the Derm7pt dataset [25].
- The `warbler` and `flycatcher` datasets are image classification tasks derived from the CalTech-UCSD Birds dataset [26] to identify different species of birds.

- The `noisyconcepts25` and `noisyconcepts75` datasets are synthetic classification tasks designed to control the noise in concepts (see Appendix B.1).

We process each dataset to binarize categorical concepts (e.g., `WingColor` to `WingColorRed`). We split each dataset into a training sample (80%, used to build a selective classification model) and a test sample (20%, used to evaluate coverage and selective accuracy in deployment).

**Models**   We train a selective classification model using one of the following methods:

- X→Y MLP: A multilayer perceptron trained on top of the penultimate layer of the embedding model. This baseline represents an end-to-end deep learning model that directly predicts the output without concepts.
- Baseline: An independent concept bottleneck model built from concept detectors $g_1, \ldots, g_m$ and a front-end model $f$ trained to predict the true concepts.
- CS: Conceptual safeguard built from the same concept detectors and front-end as the baseline. This model propagates the uncertainty from the concept detectors $g_1, \ldots, g_m$ to the front-end model $f$ as described in Section 3.

We build Baseline and CS models using the same front-end model $f$ and concept detectors $g_1, \ldots, g_m$. We train the front-end model $f$ using logistic regression, and the concept detectors using embeddings from a pre-trained model [i.e., InceptionV3, 27] (for all datasets other than `noisyconcepts`).

**Evaluation**   We report the performance of each model through an *accuracy-coverage curve* as in Fig. 2, which plots its coverage and selective accuracy on the test sample across thresholds. We evaluate the impact of confirmation in concept based models – i.e., Baseline and CS – in terms of the following policies:

- ImpactConf, which flags concepts to review using Algorithm 1;
- RandomConf, which flags a random subset of concepts to review.

We control the number of examples to confirm by setting a *confirmation budget*, and plot accuracy-coverage curves for confirmation budgets of 0/10/20/50% to show how performance changes under no/low/medium/high human levels of human intervention, respectively.

## 4.2   RESULTS

We present the accuracy coverage curves for all methods and all datasets in Table 1. Given these curves, we can evaluate the gains to uncertainty propagation by comparing Baseline to CS, and the gains from confirmation by comparing Baseline + RandomConf to CS + ImpactConf).

Overall, our results that our methods outperform their respective baselines in terms of coverage and selective accuracy. In general, we find that the gains vary across datasets and confirmation budgets. Given a desired target accuracy, for example, we achieve higher coverage on `warbler` and `flycatcher` rather than `melanoma` and `skincancer`. Such differences arise due to differences in the quality of concept annotations and their relevance for the prediction task at hand.

**On Uncertainty Propagation**   Our results highlight how uncertainty propagation can lead to improvements in a safeguard. On the one hand, we find that uncertainty propagation improves the accuracy coverage trade-off. On the `skincancer` dataset, for example, we see a major improvement in the accuracy coverage curves between a model that propagates uncertainty (CS + RandomConf, blue) and a model and a comparable model that does not (Baseline + RandomConf, green).

On the other hand, accounting for uncertainty can improve these trade-offs by producing a more effective confirmation policy. Our results highlight these effects by showing gains of uncertainty may change under a confirmation budget. On the `flycatcher` dataset, for example, accounting for uncertainty leads to little difference when we do not confirm examples (i.e., a 0% confirmation budget). In a regime where we allow for confirmation – setting a 20% confirmation budget – we find that accounting for uncertainty can increase coverage, from $46.2\%$ to $63.5\%$, when the threshold is set at $\tau = 0.05$ (see  Section 4.1).

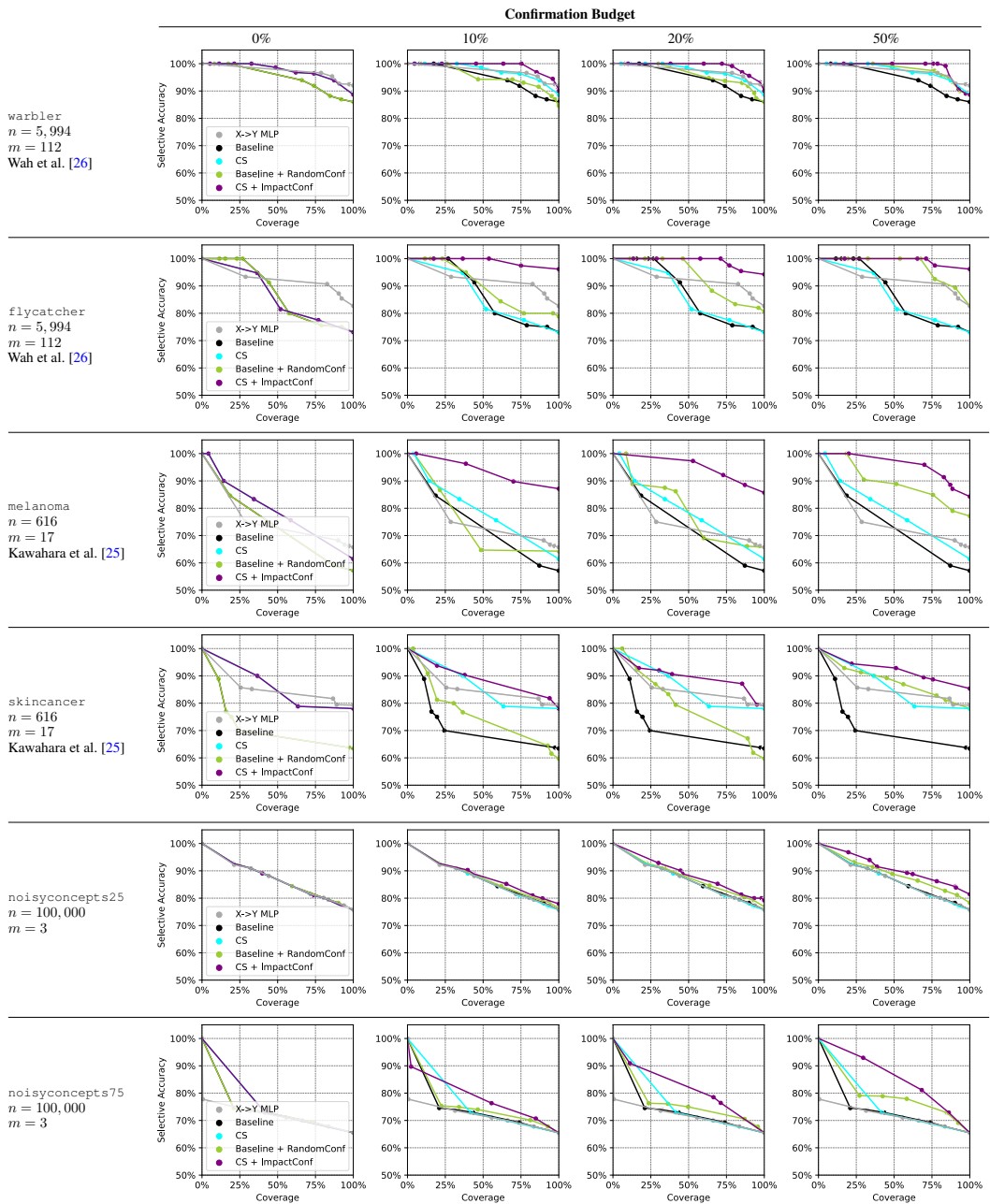

**Table 1:** Accuracy coverage curves for all methods on all datasets. We include additional results in Appendix B for multiclass tasks. Note that Baseline (black) and Baseline + RandomConf (green) produce identical results without confirmation. Likewise, CS + RandomConf (blue) and CS + ImpactConf (purple) are also equivalent under the same condition.

**On Confirmation**   Our results show that confirming concepts improves performance across all datasets. On flycatcher, we find that confirming a random subset of instances 20% improves coverage from 26.9% to 46.2% for a threshold of $\tau = 0.05$ (Baseline $\rightarrow$ Baseline + RandomConf). In a conceptual safeguard, coverage starts from 63.5% due to uncertainty propagation (CS + RandConf), and improves to 84.6% as a result of our targetted confirmation policy (CS + ImpactConf). The gains of confirmation depend on the underlying task and dataset. For example, the gains may be smaller when the concept detectors perform well enough without confirmation to achieve high accuracy (e.g., for the warbler and noisyconcepts25 datasets).

| Dataset | Prediction Thresholds | X->Y MLP | Baseline | CS | Baseline + RandomConf | CS + ImpactConf |
|---|---|---|---|---|---|---|
| `warbler` | $\tau = 0.05$ | 86.0% | 17.3% | 74.0% | 30.0% | 90.00% |
| $n = 5,994$ | $\tau = 0.1$ | 100.00% | 74.0% | 87.3% | 89.3% | 100.00% |
| $m = 112$ | $\tau = 0.15$ | 100.00% | 100.00% | 100.00% | 100.00% | 100.00% |
| Wah et al. [26] | $\tau = 0.2$ | 100.00% | 100.00% | 100.00% | 100.00% | 100.00% |
| `flycatcher` | $\tau = 0.05$ | 0.0% | 26.9% | 0.0% | 46.2% | 84.62% |
| $n = 5,994$ | $\tau = 0.1$ | 82.7% | 44.2% | 36.5% | 46.2% | 100.00% |
| $m = 112$ | $\tau = 0.15$ | 92.3% | 44.2% | 36.5% | 65.4% | 100.00% |
| Wah et al. [26] | $\tau = 0.2$ | 100.00% | 57.7% | 51.9% | 100.00% | 100.00% |
| `melanoma` | $\tau = 0.05$ | 0.0% | 0.0% | 4.3% | 8.6% | 52.86% |
| $n = 616$ | $\tau = 0.1$ | 0.0% | 0.0% | 14.3% | 8.6% | 72.86% |
| $m = 17$ | $\tau = 0.15$ | 0.0% | 0.0% | 14.3% | 41.4% | 100.00% |
| Kawahara et al. [25] | $\tau = 0.2$ | 0.0% | 18.6% | 34.3% | 41.4% | 100.00% |
| `skincancer` | $\tau = 0.05$ | 0.0% | 0.0% | 0.0% | 6.10% | 0.0% |
| $n = 616$ | $\tau = 0.1$ | 0.0% | 0.0% | 36.6% | 15.9% | 39.02% |
| $m = 17$ | $\tau = 0.15$ | 32.9% | 11.0% | 36.6% | 28.0% | 85.37% |
| Kawahara et al. [25] | $\tau = 0.2$ | 86.59% | 11.0% | 36.6% | 36.6% | 85.4% |
| `noisyconcepts25` | $\tau = 0.05$ | 0.0% | 0.0% | 0.0% | 0.0% | 0.0% |
| $n = 100,000$ | $\tau = 0.1$ | 32.3% | 32.3% | 32.3% | 33.9% | 44.66% |
| $m = 3$ | $\tau = 0.15$ | 44.1% | 43.6% | 43.6% | 45.8% | 69.16% |
| | $\tau = 0.2$ | 80.4% | 80.4% | 80.4% | 86.8% | 93.31% |
| `noisyconcepts75` | $\tau = 0.05$ | 0.0% | 0.0% | 0.0% | 0.0% | 0.0% |
| $n = 100,000$ | $\tau = 0.1$ | 0.0% | 0.0% | 0.0% | 0.0% | 11.17% |
| $m = 3$ | $\tau = 0.15$ | 0.0% | 0.0% | 0.0% | 0.0% | 11.17% |
| | $\tau = 0.2$ | 0.0% | 0.0% | 0.0% | 0.0% | 11.17% |

**Table 2:** Coverage at specific thresholds $\tau$ for a confirmation budget of 20%. We present results for other datasets and confirmation budgets in Appendix B.2.

Our results highlight the value of targetted confirmation policy – i.e., as a technique that can lead to meaningful gains in coverage without compromising safety or requiring real-time human supervision. In practice, these gains only part of the benefits of our approach – as practitioners may be able to specify costs in a way that limits the experts required for confirmation. For example, non-expert users may be able to confirm concepts such as `WingColorRed`, while confirming the final species prediction to `RedFacedCormorant` may require greater expertise.

## 5 CONCLUDING REMARKS

Conceptual safeguards reflect a general-purpose approach to promote safety through selective classification. In applications where we wish to automate routine tasks that humans could perform, safegaurds allow us to reap some of the benefits of automation through abstention in a way that can promote interpretability and improve coverage. Although our work has primarily focused on binary classification tasks, our machinery can be applied to build conceptual safeguards for multiclass tasks (see Appendix B.3), and extended to other supervised prediction problems.

Our approach has a number of overarching limitations that affect concept bottlenecks and selective classifiers. As with all models trained with concept annotations, we expect to incur some loss in performance relative to an end-to-end model when concepts are unlikely to capture all relevant information about the label from its input. In practice, this gap may be large – and we may reap the benefits of automation using a traditional selective classifier. As with other selective classification methods, we expect that abstention may exacerbate disparities in coverage or performance across subpopulations [see e.g., 28]. In our setting, it may be difficult to pin down the source of these disparities – as they may arise from concept annotations, model concepts, or interaction effects. Nevertheless, we may be able to mitigate these effects through a suitable confirmation policy.

Acknowledgments

We thank Taylor Joren, Mert Yuksekgonul, and Lily Weng for helpful discussions. This work was supported by funding from the National Science Foundation Grants IIS-2040880 and IIS-2313105, the NIH Bridge2AI Center Grant U54HG012510, and an Amazon Research Award.

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
