# A  OMITTED PROOFS

**Proposition 1.** *Suppose that $\overline{y}$ is a calibrated probability prediction for $y$. Then any selective classifier $\varphi_\tau(\overline{y})$ that abstains when $\overline{y}$ has confidence below $1 - \tau$ achieves accuracy at least $1 - \tau$.*

*Proof.* We show that on examples for which the selective classifier does not abstain, accuracy is at least $1 - \tau$. First, we use the Law of Total Probability to separate the cases where $\overline{y}$ is confident that $y = 0$ versus $y = 1$.

$$\Pr\left(y = \varphi_\tau(\overline{y}) \mid \varphi_\tau(\overline{y}) \neq \perp\right)$$
$$= \Pr\left(\overline{y} \geq 1 - \tau \mid \varphi_\tau(\overline{y}) \neq \perp\right) \Pr\left(y = 1 \mid \overline{y} \geq 1 - \tau\right)$$
$$+ \Pr\left(\overline{y} \leq \tau \mid \varphi_\tau(\overline{y}) \neq \perp\right) \Pr\left(y = 0 \mid \overline{y} \leq \tau\right)$$

Next, we use the definition of calibration, which states that $\Pr\left(y = 1 \mid \overline{y} = t\right) = t$ for all $t \in [0, 1]$, to write

$$\geq \Pr\left(\overline{y} \geq 1 - \tau \mid \varphi_\tau(\overline{y}) \neq \perp\right)(1 - \tau) + \Pr\left(\overline{y} \leq \tau \mid \varphi_\tau(\overline{y}) \neq \perp\right)(1 - \tau)$$
$$= (1 - \tau)$$

$\square$

# B  SUPPORTING MATERIAL FOR EXPERIMENTS

## B.1  DATASETS

**melanoma & skincancer**  These datasets are derived from the Derm7pt repository [25], which is de-identified and publicly available without patient information.   We preprocess the dataset by splitting the original seven categorical image annotations (pigment_network, streaks, pigmentation, regression_structures, dots_and_globules, blue_whitish_veil, vascular_structures) into seventeen binary concepts. We consider two tasks: predicting melanoma and predicting skincancer (melanoma or basal cell carcinoma). We split the validation indices in the original dataset into a validation set and a hold-out test set for evaluation. We then balance the resulting classes by downsampling the majority class. To train the concept models, we augment the original training images with random color enhancements and random flipping to obtain 10x total training images.

**warbler, flycatcher, cubspecies & cubtypes**  These datasets are derived from the CUB 2011 dataset [26]. We follow the same preprocessing described in [17]. warbler classifies birds of type warbler and flycatcher classifies birds of type flycatcher. cubspecies and cubtypes are multiclass datasets for predicting bird species and bird types, respectively. To train the concept models, we augment the original training images with random color enhancements and random flipping to obtain 10x total training images.

**noisyconcepts**  The noisyconcepts datasets are synthetic datasets that we primarily use to evaluate performance changes with respect to the quality of concept detectors. We sample the examples in these datasets $(\boldsymbol{x}_i, \boldsymbol{c}_i, y_i)$ from the following distribution:

$$\begin{aligned}
x_1, \ldots, x_5 &= \text{Bernouilli}(0.7) \\
\xi_1, \xi_2, \xi_3 &= \text{Bernouilli}(p_\xi) \\
c_1 &= \text{parity}(x_1, x_2, x_4) \oplus \xi_1 \\
c_2 &= \text{parity}(x_1, x_2, x_3) \oplus \xi_2 \\
c_3 &= \text{parity}(x_1, x_2, x_5) \oplus \xi_3 \\
p &= \text{logistic}(1.0c_1 + 2.0c_2 + 3.0c_3 - 2.0) \\
y &\sim \text{Bernoulli}(p)
\end{aligned} \tag{4}$$

The distribution in (4) includes an explicit noise parameter $p_\xi \in [0, 1]$ that we can set to control the noise in concepts. When $p_\xi = 0$, the values of $c_1, c_2, c_3$ operate as parity functions, which can only be learned through a sufficiently complex model. When $p_\xi > 0$, we inject noise into the concept labels by randomly flipping the values of $c_1, c_2, c_3$ with probability $p_\xi$. Thus, the noise parameter sets an upper bound on the accuracy of concept labels – and larger values of $p_\xi$ lead to less accurate concept detectors. In contrast to the real-world datasets, we train the concept detectors for the noisyconcepts datasets directly (i.e., without an embedding layer) by fitting multi-layer perceptron with a single hidden layer.

## B.2 Additional Experimental Results

| Dataset | Prediction Thresholds | X->Y MLP | Baseline | CS | Baseline + RandomConf | CS + ImpactConf |
|---|---|---|---|---|---|---|
| warbler | $\tau = 0.05$ | 86.00% | 17.3% | 74.0% | 26.0% | 85.3% |
| $n = 5,994$ | $\tau = 0.1$ | 100.00% | 74.0% | 87.3% | 86.7% | 100.00% |
| $m = 112$ | $\tau = 0.15$ | 100.00% | 100.00% | 100.00% | 97.3% | 100.00% |
| Wah et al. [26] | $\tau = 0.2$ | 100.00% | 100.00% | 100.00% | 100.00% | 100.00% |
| flycatcher | $\tau = 0.05$ | 0.0% | 26.9% | 0.0% | 38.5% | 100.00% |
| $n = 5,994$ | $\tau = 0.1$ | 82.7% | 44.2% | 36.5% | 38.5% | 100.00% |
| $m = 112$ | $\tau = 0.15$ | 92.3% | 44.2% | 36.5% | 38.5% | 100.00% |
| Wah et al. [26] | $\tau = 0.2$ | 100.00% | 57.7% | 51.9% | 96.2% | 100.00% |
| melanoma | $\tau = 0.05$ | 0.0% | 0.0% | 4.3% | 4.3% | 38.57% |
| $n = 616$ | $\tau = 0.1$ | 0.0% | 0.0% | 14.3% | 4.3% | 38.57% |
| $m = 17$ | $\tau = 0.15$ | 0.0% | 0.0% | 14.3% | 21.4% | 100.00% |
| Kawahara et al. [25] | $\tau = 0.2$ | 0.0% | 18.6% | 34.3% | 21.4% | 100.00% |
| skincancer | $\tau = 0.05$ | 0.0% | 0.0% | 0.0% | 3.66% | 0.0% |
| $n = 616$ | $\tau = 0.1$ | 0.0% | 0.0% | 36.6% | 13.4% | 37.80% |
| $m = 17$ | $\tau = 0.15$ | 32.9% | 11.0% | 36.6% | 13.4% | 37.80% |
| Kawahara et al. [25] | $\tau = 0.2$ | 86.6% | 11.0% | 36.6% | 30.5% | 93.90% |
| noisyconcepts25 | $\tau = 0.05$ | 0.0% | 0.0% | 0.0% | 0.0% | 0.0% |
| $n = 100,000$ | $\tau = 0.1$ | 32.3% | 32.3% | 32.3% | 33.2% | 39.77% |
| $m = 3$ | $\tau = 0.15$ | 44.1% | 43.6% | 43.6% | 44.8% | 65.35% |
|  | $\tau = 0.2$ | 80.4% | 80.4% | 80.4% | 84.00% | 82.7% |
| noisyconcepts75 | $\tau = 0.05$ | 0.0% | 0.0% | 0.0% | 0.0% | 0.0% |
| $n = 100,000$ | $\tau = 0.1$ | 0.0% | 0.0% | 0.0% | 0.0% | 0.0% |
| $m = 3$ | $\tau = 0.15$ | 0.0% | 0.0% | 0.0% | 0.0% | 2.38% |
|  | $\tau = 0.2$ | 0.0% | 0.0% | 0.0% | 0.0% | 2.38% |

**Table 3:** Coverage across abstention thresholds $\tau$ with confirmation budget 10%

| Dataset | Prediction Thresholds | X->Y MLP | Baseline | CS | Baseline + RandomConf | CS + ImpactConf |
|---|---|---|---|---|---|---|
| warbler | $\tau = 0.05$ | 86.0% | 17.3% | 74.0% | 84.7% | 86.67% |
| $n = 5,994$ | $\tau = 0.1$ | 100.00% | 74.0% | 87.3% | 93.3% | 92.7% |
| $m = 112$ | $\tau = 0.15$ | 100.00% | 100.00% | 100.00% | 100.00% | 100.00% |
| Wah et al. [26] | $\tau = 0.2$ | 100.00% | 100.00% | 100.00% | 100.00% | 100.00% |
| flycatcher | $\tau = 0.05$ | 0.0% | 26.9% | 0.0% | 67.3% | 100.00% |
| $n = 5,994$ | $\tau = 0.1$ | 82.7% | 44.2% | 36.5% | 76.9% | 100.00% |
| $m = 112$ | $\tau = 0.15$ | 92.3% | 44.2% | 36.5% | 90.4% | 100.00% |
| Wah et al. [26] | $\tau = 0.2$ | 100.00% | 57.7% | 51.9% | 100.00% | 100.00% |
| melanoma | $\tau = 0.05$ | 0.0% | 0.0% | 4.3% | 18.6% | 70.00% |
| $n = 616$ | $\tau = 0.1$ | 0.0% | 0.0% | 14.3% | 30.0% | 82.86% |
| $m = 17$ | $\tau = 0.15$ | 0.0% | 0.0% | 14.3% | 51.4% | 88.57% |
| Kawahara et al. [25] | $\tau = 0.2$ | 0.0% | 18.6% | 34.3% | 75.7% | 100.00% |
| skincancer | $\tau = 0.05$ | 0.0% | 0.0% | 0.0% | 0.0% | 0.0% |
| $n = 616$ | $\tau = 0.1$ | 0.0% | 0.0% | 36.6% | 28.0% | 51.22% |
| $m = 17$ | $\tau = 0.15$ | 32.9% | 11.0% | 36.6% | 56.1% | 100.00% |
| Kawahara et al. [25] | $\tau = 0.2$ | 86.6% | 11.0% | 36.6% | 84.1% | 100.00% |
| noisyconcepts25 | $\tau = 0.05$ | 0.0% | 0.0% | 0.0% | 0.0% | 19.75% |
| $n = 100,000$ | $\tau = 0.1$ | 32.3% | 32.3% | 32.3% | 36.0% | 38.69% |
| $m = 3$ | $\tau = 0.15$ | 44.1% | 43.6% | 43.6% | 65.6% | 78.19% |
|  | $\tau = 0.2$ | 80.4% | 80.4% | 80.4% | 91.6% | 100.00% |
| noisyconcepts75 | $\tau = 0.05$ | 0.0% | 0.0% | 0.0% | 0.0% | 0.0% |
| $n = 100,000$ | $\tau = 0.1$ | 0.0% | 0.0% | 0.0% | 0.0% | 29.55% |
| $m = 3$ | $\tau = 0.15$ | 0.0% | 0.0% | 0.0% | 0.0% | 29.55% |
|  | $\tau = 0.2$ | 0.0% | 0.0% | 0.0% | 0.0% | 68.25% |

**Table 4:** Coverage across abstention thresholds $\tau$ with confirmation budget 50%

### B.3 MULTICLASS CLASSIFICATION TASKS

In this Appendix, we briefly describe how to build conceptual safeguards for multiclass classification tasks and present experimental results for this setting.

In practice, the main requirement for adapting uncertainty propagation to cover multiple labels. In practice, this requires replacing the concept prediction vector with a matrix that encodes $\Pr(y|c)$ for all $y \in \mathcal{Y}$ and $c \in \{0,1\}^m$. For the selective classifier $\varphi_\tau$, we estimate uncertainty based on the likelihood of the most probable class and threshold prediction accordingly.

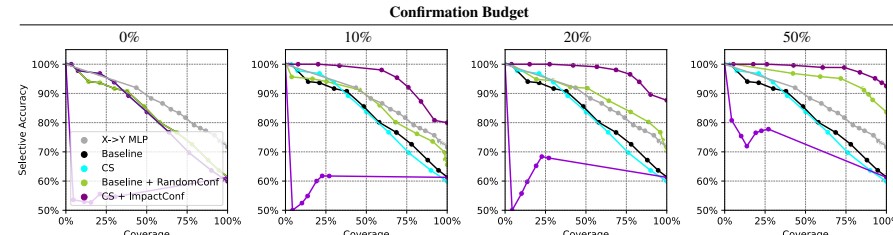

**Table 5:** Coverage vs. accuracy for all methods on multiclass classification tasks.