# OpenReview forum: "Classification with Conceptual Safeguards"
_ICLR.cc/2024/Conference — ICLR 2024 poster_

### Official Review · Reviewer_1z66 · 2023-10-27

**Soundness:** 3 good
**Presentation:** 3 good
**Contribution:** 3 good
**Rating:** 6
**Confidence:** 4

**Summary:**

The paper introduces a perspective on selective classification within deep learning using concepts. It suggests a strategy for balancing accuracy and coverage by abstaining from predictions in situations where errors can be costly. The proposed approach involves creating a concept bottleneck model, enabling the front-end model to use soft concepts and improving coverage and performance through concept confirmation. The paper presents techniques for handling uncertainty and pinpointing concepts for confirmation.

**Strengths:**

The paper exhibits good clarity in its articulation, with ideas clearly presented and structured in an organized manner. Exploring the integration of user feedback into ML models to enhance accuracy and ensure broad coverage is intriguing and holds significance for ML models in real life usage. Furthermore, the paper touches interpretability in machine learning, which is an important aspect for ML models in real life.

**Weaknesses:**

1. The abstract should be expanded to encompass key concepts that effectively summarize the paper's contributions. In the introduction, the authors emphasize the significance of interpretability and the challenges it poses in achieving high accuracy. By including these vital points in the abstract, the paper can provide a more comprehensive overview of its content and contributions.

2. Regarding the abstention process, it appears to be based on a prediction probability threshold, where if the probability is lower than the threshold, the prediction is abstained? How does it different from a decision threshold used by the models? Can authors clarify that?

3. In the results and discussion section, there's limited exploration and commentary on the impact of the solution on system accuracy, as seen in Table 2. Notably, the confirmation budget appears to have a limited effect on datasets like "noisyconcepts25" and "warbler" compared to others. The paper can delve into the reasons behind this discrepancy.

4. In real-world applications of this solution, questions about the ease of concept approval and handling conflicting user feedback arise. While these aspects may be considered out of scope, addressing them would be beneficial for evaluating the practicality of implementing this approach in real-world scenarios. This is particularly important when considering the potential challenges of user feedback and conflicting inputs in such applications.

Minor things:
Page 4, confirm. we —> replace . with comma
Section 4.2, Table Table 2 —> Table 2
Shouldn’t Table 2 rather be labelled as Figure 2?

**Questions:**

stated above

---

> ### Author Response · Authors · 2023-11-17
>
> Thank you for your feedback and thoughtful comments! We address your comments and questions below.
>
> > Regarding the abstention process, it appears to be based on a prediction probability threshold, where if the probability is lower than the threshold, the prediction is abstained? How does it different from a decision threshold used by the models?
>
> The prediction probability threshold $\tau$ functions similarly to a decision threshold in that it maps the raw output of a model to a set of outcomes. In the traditional binary classification task, a decision threshold would map a score to one of two classes (0 or 1). In our setting, $\tau$ is used to map a predicted probability to one of three classes (0, 1, or abstain).
>
> This extension is particularly significant in our setup, where we apply the abstention threshold to predicted probabilities from a calibrated classifier. In this case, the value of $\tau$ can be set so that the model abstains from a decision when the prediction probability does not meet the established confidence level. More generally,  it can be tuned to control for the coverage or selective risk (similarly to how a decision threshold can be tuned to control for TPR/FPR).
>
>
> > The abstract should be expanded to encompass key concepts that effectively summarize the paper's contributions . . . [such as] the significance of interpretability and the challenges it poses in achieving high accuracy
>
> Thank you for bringing this to our attention. We agree! We plan to revise the abstract more clearly following the discussion with the reviewers.
>
> > In the results and discussion section, there's limited exploration and commentary on the impact of the solution on system accuracy, as seen in Table 2. Notably, the confirmation budget appears to have a limited effect on datasets like "noisyconcepts25" and "warbler" compared to others.
>
> Thank you for bringing up this point!
>
> To be clear, this is indeed the correct interpretation of our results – i.e., there is a limited gain from confirmation on "noisyconcepts25" and "warbler" datasets. In practice, this can arise for several reasons.
>
> - The concept detectors perform well enough without confirmation to achieve high accuracy (this is the case for the "noisyconcepts25" and "warbler" datasets).
> - The dataset contains a large number of instances where confirmation cannot lead to gains. This can happen when, for example, instances for which we confirm concepts are inherently uncertain – i.e. so that $\textrm{Pr}(y \mid c)$ is large even when we know ground truth $c$.
> - The confirmation policy is not working well – i.e., it is somehow selecting the wrong instances for confirmation. This is not the case for the datasets that you listed. If this were the case, for example, we would expect a large gain in accuracy once we increase the confirmation budget from 75% to 100%.
>
> We’ve added these points to our discussion in Section 4.
>
> > In real-world applications of this solution, questions about the ease of concept approval and handling conflicting user feedback arise. While these aspects may be considered out of scope, addressing them would be beneficial for evaluating the practicality of implementing this approach in real-world scenarios.
>
> Thank you for bringing up this point.
>
> In our approach, we primarily focus on what we term "deterministic concepts." These are concepts where there is a high degree of consensus among informed individuals. For instance, in medical imaging, the identification of a bone spur on an X-ray is generally agreed upon by radiologists. Among non-expert users, identifying that a bird has a red wing (rather than a black wing) is likely to lead to consensus. These examples represent areas where expertise leads to a high degree of agreement. It's crucial to note that it's not necessary for every human to agree; rather, the consensus should exist among those with relevant expertise. For example, in medical applications, we would rely on the agreement among medical professionals. By focusing on such concepts, we can set appropriate controls for concept selection and confirmation, effectively managing the costs and complexities involved. We plan to include a comprehensive list and categorization of these concepts in our paper to enhance clarity and facilitate a more practical evaluation of our approach in real-world scenarios.
>
>
> > Minor things
>
> Thank you for pointing these out! We’ve made corrections in the updated version of the paper according to your comments. We’ve left Table 2 as a Table of figures rather than a Figure, though we appreciate your comment that this may be confusing.

---

### Official Review · Reviewer_msQs · 2023-10-30

**Soundness:** 3 good
**Presentation:** 4 excellent
**Contribution:** 3 good
**Rating:** 8
**Confidence:** 3

**Summary:**

The authors propose a classification system that uses the combined approaches of a conceptual bottleneck and abstaining outputs to increase the reliability of models. The conceptual bottleneck approach trains a classification model for each concept identified in the training data. The end-model is a classifier that uses the presence or absence of concepts to make the target classification. The abstain mechanism allows the end-model to abstain from prediction.  When the model is uncertain about the presence of a concept, it may query the user for confirmation, thereby increasing trust and performance. Concept uncertainies are propagated through the end-model by using concept identification model scores as probabilities and sampling over potential concept vectors. This also improves performance.

**Strengths:**

The three strengths of the proposed approach are a functional abstaining method, requests for confirmation, and uncertainty propagation. Together these methods raise a classification model to something that is more intelligent, capable of some corrective action when faced with unusual inputs.

**Weaknesses:**

1. The uncertainty propagation methodology doesn't seem computationally efficient.
2. The performance of the default classifier (always predict majority class, uniformly randomly abstain) ought to be included in Table 2. The default performance ought to always be presented when using accuracy as a performance metric.

**Questions:**

Can a deeper analysis of the consequences of abstaining be provided? Abstaining almost always improves average performance on the remaining predictions. Reporting the average is almost illusory, since those non-abstain predictions would have been correct or incorrect regardless. Rather, there is a real cost associated with refusing to provide an answer. The benefit is that the model reduces risk of error, but the costs are application dependent. How can we think about these costs in a constructive manner?

---

> ### Author Response · Authors · 2023-11-17
>
> Thank you for your time and thoughtful comments! We address your comments and questions below and included an updated version with additional experiments.
>
> > The performance of the default classifier (always predict majority class, uniformly randomly abstain) ought to be included in Table 2
>
> Thank you for pointing this out! We agree. In response, we have included the coverage-accuracy curve of a baseline model that employs uniform random abstention in Table 2's results. This baseline is designated as "Random" in the table's legend and is represented by a dark gray line in the corresponding plots. Additionally, we intend to incorporate a baseline that predicts the majority class into the supplementary results on multiclass datasets presented in Appendix A.3 in our next revision.
>
> > The uncertainty propagation methodology doesn't seem computationally efficient.
>
> This is true – the uncertainty propagation method presented in section 3.1 requires $2^m$ calls to the front-end model, where $m$ is the number of concepts. However, in practice, Monte Carlo sampling can be used whenever the number of concepts is larger than is computationally feasible for exact calculation. In our experiments, we found that the Monte Carlo estimation converged quickly to the true value for larger datasets, and were able to run experiments across all datasets in a matter of minutes using a single GPU. Thank you for pointing out that this may not be clear on an initial read – we’ve updated section 4.1 accordingly.
>
> > There is a real cost associated with refusing to provide an answer. The benefit is that the model reduces risk of error, but the costs are application-dependent. How can we think about these costs in a constructive manner?
>
> This is a great point and actually part of the reason why we focus on "automation" as a motivating class of applications – i.e., applications where we would build a deep learning model to automate a routine task that a human expert can perform, like detecting pneumonia from a chest x-ray or melanoma from an image of a skin lesion.
>
> In these applications, there is a constructive role for abstention for two reasons. First, accuracy is a major bottleneck to deploying any predictive model - i.e., even a model that achieves a 10% error rate may not be accurate enough to deploy in a medical setting. Second, there is a clear baseline option for what to do when we abstain. If we did not use a predictive model, then we can continue to have medical professionals perform routine yet important tasks.
>
> With this in mind, we think that the right framing is to associate a “gain” with any prediction rather than a “cost” with an abstention. Put differently, using existing methods, we may not be able to automate at all. Using abstention, however, we can automate at least some of the instances. This is also the broader case that surrounds some recent work on automation - see, e.g., Feng, Jean, et al. "Selective prediction-set models with coverage guarantees." Biometrics 79.2 (2023): 811-825.
>
> In our manuscript, we follow a “bounded risk” setup for selective classification where practitioners seek to build a system that will maximize coverage while ensuring that its performance exceeds a desired level of accuracy. In other words, we seek to minimize the costs associated with abstention. In comparison to traditional selective classification, the one benefit of our approach is that we can “confirm” to further improve coverage.  An alternative setup for selective classification is to set fixed costs associated with misclassification and with abstention and to minimize their additive costs. We think this is also a valid setup, but note that in many applications it is difficult to specify these quantities.
>
> We hope that this provides some answer to your question. We plan to express this more clearly in our manuscript and are happy to discuss more if needed.

---

> > ### Comment · Reviewer_msQs · 2023-11-22
> > **Author responses**
> >
> > I thank the authors for their responses and hard work. I will stand by my score.

---

### Official Review · Reviewer_3F5i · 2023-11-01

**Soundness:** 3 good
**Presentation:** 3 good
**Contribution:** 3 good
**Rating:** 8
**Confidence:** 3

**Summary:**

The authors propose the use of a concept bottleneck model as input for selective classification. Moreover, they propose a greedy algorithm to select concepts to be confirmed by human experts, with the objective to increase the coverage of the selective classifier while guaranteeing a minimum accuracy level of the selective classifier. They evaluate their method with competitive baselines using both synthetic and real datasets.

**Strengths:**

The work appears to be the first to use concept bottleneck models to capture the uncertainty of the entire model for selective classification. Moreover, the idea of getting human feedback to confirm concepts to improve selective classification is quite interesting and adds to the increasing literature of human-in-the-loop algorithms.

The paper is very well organized, has a clear structure, and is nicely written. The authors clearly state their contributions as well as the assumptions of their method. They also provide a detailed description of the experimental setup and provide the code for reproducibility in an anonymized repo. The experimental evaluation seems comprehensive including experiments with on both synthetic and real datasets, as well as a  robustness analysis under violations of the Assumptions 1 and 2.

**Weaknesses:**

Even though the meaning of coverage might be clear to experts in selective classification, it might be helpful to include a high level definition of coverage in the introduction, so that it is clear for a broader ML audience.

In Proposition 4, the authors assume a perfectly calibrated predictor. However, in practice, perfect calibration is impossible. As a results, it would be useful to include theoretical results that complement proposition 4 that account for the calibration error a classifier.

Style/Typos:
1. Figure 3 has no caption.
2. The style of citations and captions of tables and figures does not follow the ICRL author instructions.

**Questions:**

1. Assuming that there is (small) calibration error of the predictions of the classifier, how would the results of proposition 4 change?
2. It seems that abstention happens when $\bar{y}_i = \tau$. Could one also assume that abstention happens when $\bar{y}_i \in( \tau_1, \tau_2)$, that is when the prediction of the classifier is within some range? How could this affect the results of proposition 4, as well as the accuracy guarantees assuming a not perfectly calibrated classifier?

---

> ### Author Response · Authors · 2023-11-17
>
> Thank you for your time and feedback! We address your comments and questions individually below.
>
> > It would be useful to include theoretical results that complement proposition 4 that account for the calibration error a classifier . . . Assuming that there is (small) calibration error of the predictions of the classifier, how would the results of proposition 4 change?
>
> This is an important point. In short, you are right that Proposition 4 assumes that the front-end model will output calibrated probability predictions. In practice, this is a reasonable assumption as we can usually build such models using standard post-hoc calibration techniques. In the event that the model is not calibrated, then Proposition 4 will hold only given the degree of calibration. For example, given a miscalibrated model such that $|Pr(y = 1 | p = t) - t| = \epsilon > 0$ for some error rate $\epsilon$, the accuracy in Proposition 4 is guaranteed to be $1 - \tau - \epsilon$ instead of $1 - \tau$.
>
> We can address this case by setting $\tau$ using a more sophisticated method that works with “confidence scores” that may or may not be calibrated. One such example is the “Selection with Guaranteed Risk Control” algorithm by Geifman and El-Yaniv [1], which can set threshold $\tau$ so that the accuracy of the resulting system will exceed accuracy $r*$ with probability $1-\delta$.
>
> In both cases, we can obtain theoretical guarantees on the selective risk of the system under the standard PAC model. These guarantees would let us say that the accuracy of our system will surpass a user-specified threshold on accuracy when we set the abstention threshold to $\tau$.
>
> We’ve included this discussion in our revision.
>
>
> > It seems that abstention happens when $\bar{y}_i = \tau$.
> Could one also assume that abstention happens when $\bar{y}_i = (\tau_1, \tau_2)$?  That is when the prediction of the classifier is within some range?
>
> This is actually the correct assumption. To be clear, given \bar{y}_i we would operate as follows:
>
> - $\bar{y}_i > \tau \implies$ predict ($\hat{y}_i = 0$)
> - $\bar{y}_i \in (\tau, 1-\tau) \implies$ abstain ($\hat{y}_i = \perp$)
> - $\bar{y}_i < \tau \implies$ predict ($\hat{y}_i = 1$)
>
>
> > It might be helpful to include a high-level definition of coverage in the introduction
>
> Thanks for raising this point. We use coverage to denote the proportion of samples for which a predictor outputs a prediction rather than abstaining. In other words, given a model $h$, we define coverage as
> $\mathrm{Coverage}(h) := \mathrm{Pr}(\hat{y}\neq\perp)$. In our problem setting, we seek to improve coverage by maximizing the proportion of samples on which the system can safely predict. We’ve added these notes to the introduction.
>
>
> > Figure 3 has no caption.
>
> Thanks for catching this! It looks like a compilation error. We’ve fixed this in the latest manuscript.
>
> > Citation and Caption Style
>
> Thanks for catching this as well! We’ve adapted in the latest version.
>
> [1] Y Geifman, R El-Yaniv, Selective Classification for Deep Neural Networks NeurIPS 2017

---

> > ### Comment · Reviewer_3F5i · 2023-11-22
> >
> > I would like to thank the authors for taking the time to answer my questions and revise the manuscript to address any issues.

---

### Official Review · Reviewer_wCU7 · 2023-11-02

**Soundness:** 2 fair
**Presentation:** 2 fair
**Contribution:** 2 fair
**Rating:** 6
**Confidence:** 3

**Summary:**

The authors present an approach to do selective classification in deep learning with concepts, by constructing a concept bottleneck model where the front end model can make predictions given soft concepts and leverage concept confirmations to improve coverage and performance under abstention.

**Strengths:**

The authors provide a good motivation and introduction. Authors also provide emperical validations on multiple datasets. The problem statement is very relevant to practical problems and provide an insight into how to automate classification tasks by making it safe and interpretable.

**Weaknesses:**

The writing and flow could be improved better, some of them are raised in questions below. Table 1 is referenced in Section 1, however what the columns means is defined only in Section 2, which makes it harder to read the table meaning.

It would also be better to provide more details in the evaluation dataset around what each datasets means, and some statistics around it.

In my opinion the paper lacks novelty in terms of the innovation, and answers to the questions raised would help to understand better. Its not very clear about dataset statistics and how it changes and aligns with the interpretations that are presented.

**Questions:**

In the introduction its mentioned the front model can make predictions given soft concepts, however later in the text its mentioned in Section 2 under: `Propagating Concept Uncertaininty` its mentioned the front-end model requires hard concepts as inputs, which is not very clear?

In Introduction, its not very clear why the two objectives would conflict with other, if there are papers to cite that would help to make the claim stronger?

How does the choice of models to more complex architectures change the performance of the system?

---

> ### Author Response · Authors · 2023-11-17
>
> Thank you for your time and feedback! We respond to your comments below and have included an updated version of the paper with additional experiments.
>
> > How does the choice of models to more complex architectures change the performance of the system?
>
> This is a good question. Based on your question, we ran ablation studies where we swapped the logistic regression concept detectors for multi-layer perceptron concept detectors and have included these results in the updated version of the paper under Appendix A.2. We found that using a more complex model can improve baseline performance, though improvements are task-dependent, and usually small compared to strategies like improving the confirmation policy or increasing the confirmation budget. For example, using an MLP instead of logistic regression improves performance across all strategies on the "flycatcher" dataset, with the greatest gains seen in the Baseline strategy. However, this improvement is small compared to increasing the confirmation budget from even 10\% to 20\%. Stepping back, one of the benefits of our proposed approach is that it provides some flexibility over the architecture used for concept detectors and the front-end model.
>
> > In the introduction it’s mentioned the front model can make predictions given soft concepts, however later . . . [it’s] mentioned the front-end model requires hard concepts as inputs, which is not very clear?
>
> We can see how this may have been unclear and have updated Figure 1 and Section 2 to clarify this point. To be clear, traditional concept bottleneck models are designed to use hard concepts, and in conceptual safeguards, the front-end model is still trained on hard labels. One key innovation of conceptual safeguards is the ability to use soft labels at inference time using the uncertainty propagation technique in Section 3.
>
> > In Introduction, its not very clear why the two objectives would conflict with other, if there are papers to cite that would help to make the claim stronger?
>
> Thank you for bringing this to our attention. We are specifically discussing this conflict in the context of concept bottleneck models. In this case, we can substantially improve interpretability over traditional deep learning models because we can show how the model makes its predictions in terms of concepts. At the same time, this architecture leads to a loss of accuracy since datasets may not have a large number of useful concepts [see 1,2]. We’ve rephrased the introduction and added these citations to make this more clear.
>
> > It would also be better to provide more details in the evaluation dataset around what each datasets means, and some statistics around it.
>
> We included details on the datasets, including the source of the dataset, the task outcome description, the number of samples, the number of concepts, and the number of positive samples in Appendix A.1. Could you take a look and let us know if there anything else you’d like to see?
>
> > Table 1 is referenced in Section 1, however what the columns means is defined only in Section 2
>
> Thank you for raising this point – we’ve adjusted the flow of the text to make this easier to read.
>
> [1] M Yuksekgonul et al. Post-hoc concept bottleneck models
>
> [2] C Yeh et al. On completeness-aware concept-based explanations in deep neural networks

---

> ### Comment · Reviewer_wCU7 · 2023-11-22
> **Changing my assessment to accept**
>
> Thanks authors for providing a detailed review. Having read the response to my questions and to other questions. I will udpate my assessment !

---

### Author Response · Authors · 2023-11-17

We thank all reviewers for their time and feedback!

We were excited to see that reviewers described our work as “very relevant to practical problems” (wCU7) and acknowledged that our work “holds significance for ML models in real life usage” (1z66) and “adds to the increasing literature of human-in-the-loop algorithms” (3F5i). We appreciate that reviewers understood the motivation of this work, noting it “provides an insight into how to automate classification tasks by making it safe and interpretable” (wCU7) and allows for a “classification model . . . that is more intelligent, capable of some corrective action” (msQs).

We have addressed questions and comments in our responses to each reviewer. Key updates in the revised paper include: (i) additional experiments to evaluate the impact of complex architectures, now included in Appendix A.2; (ii) inclusion of an additional baseline in Table 2 for uniformly random abstention; (iii) an enhanced explanation of the uncertainty propagation mechanism in Section 2 and Figure 1; (iv) expanded discussion on the effects of conceptual safeguards on task accuracy in various contexts.

Thank you again for your time and engagement. We look forward to answering any other questions you may have during the discussion period.

---

### Meta-Review · Area_Chair_KXnP · 2023-12-14

**Metareview:**

The work proposes a sequential image classification pipeline that, given the input image, first, infers the concepts, then uses them to predict the classification labels and confidence scores, and finally then decides whether to keep this previously generated prediction or not (i.e. abstain). The literature review is thorough.

The work is the first to use concept bottleneck models in "selective classification" i.e. deciding whether to predict or abstain. The method is tested thoroughly on an array of diverse datasets (bird to medical images). Most reviewers found the work to be practical, interesting and relevant to the future human-AI interaction and human-in-the-loop frameworks. The manuscript also clearly states the limitations of this work.

The paper received positive feedback from all reviewers.   The AC recommends `Accept`.

**Justification For Why Not Higher Score:**

The work has not been demonstrated on large-scale datasets for example ImageNet.
And it requires specific, well-defined concepts for each tested dataset.

**Justification For Why Not Lower Score:**

The work tested on diverse datasets and is novel.

---

### Decision · Program_Chairs · 2024-01-16

Accept (poster)